# An Analysis of the Pass-Through of Exchange Rates in Forest Product Markets

**Selin Güney** [1,*]🔹, **Andrés Riquelme** [2]🔹 and **Barry Goodwin** [3]

1    Department of Agriculture Education and Communication, Tarleton State University, Stephenville, TX 76402, USA
2    Faculty of Economics and Business, University of Talca, Talca, Maule 3460000, Chile; juriquelme@utalca.cl
3    Department of Agricultural and Resource Economics, North Carolina State University, Raleigh, NC 27695, USA; barry_goodwin@ncsu.edu
*    Correspondence: sguney@tarleton.edu

**Abstract:** This paper assesses the exchange rate pass-through (ERPT) for forest product prices (i.e., sawnwood, logs) by applying a two-regime Self-Exciting Threshold Autoregressive (SETAR) model. We incorporate autoregressive second-order dynamics in the regime equations. This leads to better forecasts, as integrating more lags helps capture the cumulative effects of the price dynamics. We examine sawnwood and log products traded in the United States, Malaysia (Southeast Asia) and Cameroon (West Africa). Our results illustrate the importance of applying the two-regime SETAR-type models to analyze the non-linear exchange rate pass-through for forest product markets. The impulse response analysis of each price pair supports the changing behavior of price ratios in various regimes. This may be regarded as another justification to apply models accounting for structural changes to investigate the exchange rate pass-through in a non-linear fashion. The aftershock adjustment process is similar, but the amplitude of the impact differs among markets. The results reveal potential arbitrage opportunities in the forestry industry.

**Keywords:** two-regime SETAR; exchange rate pass-through; forest product market; international trade

**JEL Classification:** $Q_{02}$, $Q_{17}$, $Q_{23}$, $F_{14}$

## 1. Introduction

Forest product trade is an important source of foreign exchange earnings for many developing countries, particularly those in Southeast Asia and West Africa. Two of the largest tropical rain forests in the world are located in West and Central Africa. These two rain forests contain nearly 230 million hectares of forest and more than 300 different tree species. Consequently, the region is an influential producer of wood products. Wood production in the region is highly dominated by industrial round wood (e.g., sawnwood, wood-based panels). Historically, since the early 1990s, West and Central Africa have been exporters of logs [1]. Most of the local demand is for low-grade timber used in regional construction projects. High-grade timber is exported to the European Union (i.e., France, Germany), China and the United States (US) [2]. In 2021, Cameroon exported USD 151 million worth of industrial round wood that led the country to be the top exporter in Africa [1].

As a result of the increasing population, urbanization and the development of the middle class in Sub-Saharan Africa [3], the regional demand for wood products used for building materials and furniture is rapidly increasing. According to ITTO's Biennial Review and the Assessment of the World Timber Situation Report [4], the Central African Republic log production was 554,000 m$^3$ in 2020. Log exports, which were only 114,000 m$^3$ in 2014, had risen to 418,000 m$^3$ in 2018. In 2020, log exports amounted to 196,000 m$^3$. Sawnwood production was estimated by ITTO to be only 41,000 m$^3$ in 2020. Sawnwood exports were estimated by ITTO at only 13,800 in 2020, up from 12,000 m$^3$ in 2018 and down from

15,000 m$^3$ the previous year. Export value of timber products from the Central African Republic increased to USD 69.2 million in 2020 from USD 52.7 million in 2018, driven mainly by rising trade in logs destined for China [4]. Between 2018 and 2021, the timber sector expanded, resulting in regional development and new job opportunities.

Malaysia is one of the world's largest exporters of tropical timber products [5]. In 2021, Malaysia produced 3.05 million m$^3$ of logs, of which 1.19 million were exported; these exports were worth USD 570.3 million [6]. In 2021, the forestry and logging industry contributed 0.4 percent to the gross domestic product (GDP) of Malaysia [7]. The major export market destinations for Malaysia's wood products are Japan, US, India, Singapore and China [8].

In the US, more than 303 million acres and about two thirds of all US forested land is timberland. These resources contribute to a strong forest products industry. The forest sector is among the most important employers in the USA with approximately 953,000 employees [9]. The US industrial roundwood sector produces lumber, pulp products, plywood and veneer [9]. According to the USDA Foreign Agriculture Service (FAS)'s 2022 year-end statistics, there was a 7 percent increase in forest product exports between 2021 and 2022. In 2021, US forest product imports were valued at USD 34 billion, compared to USD 19 billion in 2017. Import values increased by 3 percent between 2021 and 2022 [10].

In developing nations, there is often a likelihood of macroeconomic instability leading to fluctuating exchange rates. One key connection between primary commodity markets and macroeconomic disruptions is how commodity prices respond to changes in exchange rates and international price shocks. Effective arbitrage in commodities and foreign exchange should ensure that once the prices for a homogeneous commodity are expressed in a standard currency, any shocks will result in balancing adjustments to keep zero profit conditions.

The market price dynamics literature is primarily based on the concept of the Law of One Price (LOP) and Purchasing Power Parity (PPP). The focus on prices and exchange rates was not only to verify if the LOP/PPP holds, but also to evaluate the impact of exchange rate changes on a nation's trade balance. There was particular interest in how a devaluation of a country's currency would affect its trade balance. It is important to determine how the devaluation would affect export prices. It is also important to determine whether or not the full effect of the devaluation would be passed through the local currency prices, especially for the importer. This led to the concept of the exchange rate pass-through (ERPT). ERPT is defined as the "percentage change in local currency import prices resulting from a one percent change in the exchange rate between the importing and exporting countries" [11,12].

The ERPT has important implications for economists and traders because it provides spatial arbitrage opportunities. This may also help policymakers determine which trade policies to impose. The underlying idea behind this concept is that the prices for identical products in various geographical areas should not vary more than the cost of transportation and transactions (such as insurance and contract fees).

In efficiently connected global markets, exchange rate shocks and price fluctuations should precisely reflect the adjustments necessary to maintain an efficient price equilibrium. Although efficiently linked markets dictate this balance, it is often observed that the pass-through effects of exchange rate shocks are not complete. As a result, standard arbitrage behavior may not be supported by empirical evidence, leading to under- or over-adjustment of prices.

Anderl and Caporale [11], Alola, Usman and Alola [13], Caselly and Roitman [14], Durmaz, Nazif, and John Kagochi [15], Goodwin et al. [16] and Wiseman, Luckstead and Durand–Morat [17] illustrate that prices will adjust in a non-linear manner. The impossibility of negative storage leads to non-linear price relationships between forest market commodities [18,19]. In addition, Lewandrowski et al. [20] found non-linearly linked prices and beginning-period inventories for the US softwood lumber sector.

Non-linear models have evolved to consider structural changes and regime switching behavior, representing the effects of transaction costs and other frictions, as well as government

policies, which may inhibit price adjustments. The existence of these frictions is often viewed as a trait that displays the overall performance of a market and is especially important as an indicator in developing and transitioning countries. Examples include Milas and Legrenzi [21]; Caselly and Roitman [14]; Wiseman, Luckstead and Durand–Morat [17]; Adewuyi, Ogebe and Oshota [22]; Alola, Usman and Alola [13]; and Fan, Talavera and Tran [23], who studied the asymmetric adjustment of real exchange rates. Further, Holmes [24] provided new insights into the nature of ERPT modeling in the context of a Markov regime-switching environment.

One of the most commonly used models that considers structural changes and regime behavior is the non-linear Smooth Transition Regression (STR) (Anderl and Caporale [11], Teräsvirta [25,26], Bhat, Nain and Bhat [27], Cheikh and Younes [28], Shintani et al. [29]). STR models are generalized threshold models with a continuous transition function. They differ from regular threshold models by allowing for smooth changes during the transition phase [11,27,30]. Hence, these models are regime dependent, in which the switches between regimes are produced endogenously (Smooth Transition Regression models are discussed in Nogueira and León–Ledesma [31], Cheikh [32], Shintani et al. [29], Wu et al. [33], Goodwin et al. [16], Haggan and Ozaki [34], Tsay [35], Granger and Teräsvirta [36], and Tong [37]). STR models have been extended to Self-Exciting Threshold Autoregressive (SETAR) models, which allow for the use of the lagged dependent variable as the regime switching driver.

Asymmetries and non-linearities are important features in exploring ERPT effects in import prices. However, few studies consider these features using modern time series methods focusing on the forestry industry at the commodity level. Baharumshah and Habibullah [38] studied the efficiency of the Malaysian foreign exchange market. Bolkesjo and Buongiorno [39] investigated the elasticity of US forest product imports and exports. Hänninen et al. [40] analyzed the long-run exchange rate elasticity of Finnish paper and pulp export prices. Tao, Diao and Cheng [19] studied the dynamic impact of Covid 19 on log prices and assessed the speed of the prices' recovery under structural changes (other studies include Hänninen et al. [41], Powers and Riker [42], Tutueanu [43] and Goodwin et al. [16]).

We follow the current literature addressing non-linearity and asymmetry issues in the forestry industry at the commodity level. In particular, the ERPT for two important timber commodities are considered: sawnwood (hard and soft) and logs (hard and soft). The trading regions considered in the analysis include the US, Malaysia (Southeast Asia) and Cameroon (West Africa). These regions have not been extensively considered in the previous literature, with the notable exceptions of Terheggen [44], Güney [45] and Wiseman, Luckstead and Durand–Morat [17].

Price dynamics will be investigated by applying two-regime SETAR models. In a smooth transition model, the patterns of price adjustment in markets are continuous rather than abrupt, even though the economic behavior driving the adjustments is discontinuous (i.e., arbitrage is either profitable or not). Likewise, asymmetric exchange rate adjustments play an important role in the exchange rate pass-through literature. The exchange rate appreciation and depreciation will have different market effects. For instance, when exporters convert their revenue earned in a foreign currency into their home currency, they have to pay a transaction cost. This cost depends on the magnitude of the exchange rate movements.

Among the papers that find evidence of commodity prices displaying regime-dependent adjustments are Balagtas and Holt [46] and Enders and Holt [47] (additional studies include Goodwin et al. [16], Enders and Holt [47], Wu et al. [33], Cheikh et al. [48] and Goodwin et al. [49]).

The primary research objective of this paper is to determine the responsiveness of international forest product prices to exchange rate variations. Another objective is to test whether or not differences in profitable arbitrage opportunities exist between regions. To do so, we estimated a two–regime SETAR model to discuss the magnitude and significance of the threshold parameters as well as the speed of adjustment that are obtained from the



estimated models. The significance of the threshold and adjustment parameters confirm the existence of arbitrage opportunities between the regions.

This paper contributes to the literature by including autoregressive second-order dynamics in the regime equations. This results in better forecasts, since integrating more lags helps capture the cumulative effects of the price dynamics. To the best of our knowledge, the current literature uses two regimes with one lag in each regime. Our approach is more flexible and allows for the incorporation of up to two lags, as well as different numbers of lags in each regime. This investigation also contributes to the time series literature focusing on the forestry industry at the commodity level by investigating forestry industry regions that have not been extensively investigated previously.

## 2. Materials and Methods

A generic regression model is used to assess the long-run price relationship, as recommended in Goldberg and Knetter [12]:

$$P_{it} = P_{jt}^{\delta} E_t^{\eta} \qquad t = 1, \ldots, T \tag{1}$$

where $P_{it}$ is the nominal price in domestic currency in region $i$ at time $t$. $E_t$ is the exchange rate between region $i$ and region $j$ (expressed as country $i$'s currency per 1 unit of country $j$'s currency). $P_{jt}$ is the nominal price in region $j$. $\delta > 0$ and $\eta > 0$ are parameters to be estimated. ERPT is complete when $\eta = 1$. Here, the effect of the exchange rate is completely passed onto the import price. ERPT exhibits an incomplete adjustment process if $\eta < 1$ and overshoots if $\eta > 1$ [50].

By taking the log of Equation (1) and adding an error term $\varepsilon_{it}$, we obtain

$$p_{it} = \delta p_{jt} + \eta e_{it} + \varepsilon_{it} \qquad t = 1, \ldots, T \tag{2}$$

where $\varepsilon_{it} \sim \text{iid}(0, \sigma^2)$. The lowercase form indicates that the variables are in logarithmic form. Following Goodwin et al. [16], we express the prices in the $i^{\text{th}}$ country's currency by expressing the price as $P_{jt} = \tilde{P}_{jt}/E_t$. Equation (1) can now be written as

$$p_{it} = (\eta - \delta)e_{it} + \delta \tilde{p}_{jt} + \varepsilon_{it} \tag{3}$$

There are several arguments in favor of the ERPT being regime-dependent. This is especially the case if the effect of the exchange rates on import prices changes with the direction and magnitude of the shocks given to the prices. Transaction costs may be involved due to the conversion of a foreign currency into a domestic currency.

Consequently, the model can be further modified as follows:

$$p_{it} = (\eta - \delta_1 \mathbb{1}_{\{S_t > c\}} - \delta_2 \mathbb{1}_{\{S_t \le c\}})e_{it} + \delta \tilde{p}_{jt} + \varepsilon_{it} \tag{4}$$

where $\mathbb{1}_{\{\cdot\}}$ is the indicator function that separates the effect of the exchange rate into two regimes. This is characterized by the state variable $S_t$ being above or below the threshold $c$ during period $t$.

For the econometric specification, we apply the 2-regime SETAR model proposed by Franses and van Dijk [51]. More specifically, our "full" model is as follows:

$$\begin{aligned} y_t = (\phi_0^{R_1} + \phi_1^{R_1} y_{t-1} + \phi_2^{R_1} y_{t-2})\, F(y_{t-1}; \gamma, \alpha, c) + \\ (\phi_0^{R_2} + \phi_1^{R_2} y_{t-1} + \phi_2^{R_2} y_{t-2})\, (1 - F(y_{t-1}; \gamma, \alpha, c)) + \varepsilon_t. \end{aligned} \tag{5}$$

where the basic unit of analysis $y_t$ is the natural logarithm of the price ratio $(p_{it}/p_{jt})$.

Equation (5) includes two regimes. The equations for the first and second regimes are

$$\phi_0^{R_1} + \phi_1^{R_1} y_{t-1} + \phi_2^{R_1} y_{t-2} \quad \text{and} \quad \phi_0^{R_2} + \phi_1^{R_2} y_{t-1} + \phi_2^{R_2} y_{t-2}$$

where $y_t$ is the dependent variable at time $t$, and $R_1$ and $R_2$ indicate the parameters corresponding to the first and second regimes, respectively. $\varepsilon_t$ is a non-identical, independent random error, with mean zero and constant variance $\sigma^2$. $F(y_{t-1}; \gamma, \alpha, c)$ is the logistic transition function (alternatively, the exponential form $F(y_{t-1}; \gamma, \alpha, c) = 1 - \exp(-\gamma(\alpha y_{t-1} - c)^2)$ can be used. We prefer the logistic form applied in Buncic [52].):

$$F(y_{t-1}; \gamma, \alpha, c) = (1 + \exp(-\gamma(\alpha y_{t-1} - c)))^{-1}, \qquad \gamma > 0. \tag{6}$$

$F(y_{t-1}; \gamma, \alpha, c)$ varies between zero and one in a smooth manner, i.e., the changes between regimes are not abrupt. The magnitude of the change depends on the transition variable $S_t = y_{t-1}$. Thus, the model is self-exciting. The characteristics of the transition function are determined by the speed of the adjustment parameter $\gamma > 0$ and the threshold parameter $c$. More specifically, if $\gamma$ is large, the transition function switches between zero and one more quickly than in the case where $\gamma$ is small. When $\gamma \to \infty$, the transition function becomes binary. For identification purposes, we set $\alpha = 1$ (see Franses and van Dijk [51] for more detail). Each distinct value of the transition function leads to a different regime.

The attributes of the transition are determined by the threshold variable $c$. We expect to observe greater differences in prices as c increases (in absolute value). This implies larger divergences from parity conditions, which indicates higher potential gains from arbitrage. Larger divergences will also result in faster and/or larger market corrections [49].

*Data*

The data cover a set of monthly average prices and exchange rates between January 1990 and November 2019. The price data are expressed in USD per cubic meter. Foreign and domestic prices were collected from the United Nations Conference on Trade and Development (UNCTAD), IndexMundi and the World Bank. Exchange rates were collected from the International Financial Statistics (IFS) curated by the International Monetary Fund (IMF).

## 3. Results

Five price series were used in the analysis: logs Cameroon, logs Malaysia, logs US, sawnwood Malaysia and sawnwood US. The price series are depicted in levels and logs in Figure 1. To investigate the ERPT theory, we created price ratios of the same commodities between different countries. Therefore, in our analysis, we accommodate four price ratios. Figure 1 provides important information about the economic conditions. A clear relationship between the prices in each market is observed. Forest product industries experienced difficult times during the early 1990s in the US. As a result of the economic recession (1989–1991), many processing plants were closed. The increasing prices during the recession helped some large firms that had private access to forest products (e.g., plywood prices increased by 65% between 1991 and 1993) [53].

The state of the general economic conditions is particularly important, given that residential construction (that directly affects wood products) demand is influenced by economic cycles. Housing sector activity is a prime indicator of the general economic conditions (Leamer [54], Stock and Watson [55], Weinstock [56]). It is not surprising that the ERPT could change with the economic activity when we consider price responses for commodities used in house construction (e.g., sawnwood, plywood).

Significant improvements in US economic activity were observed during the first half of 2002. This was an indication that the economic recession in the US was starting to end in March 2001. Despite GDP growth decreasing in early 2003, monetary policy and a strong housing sector helped the US economy recover later that year. The housing construction sector was reinforced by low mortgage rates. As a result, the US observed high demand for wood products in 2003 [57].

The price pairs reflect the effects of the 2005–2006 mortgage crisis in the US. Prices in the housing sector dropped by almost 30 percent between 2006 and 2009 and remained unchanged until March 2013 [58]. Wood product prices were hugely affected by the crisis.

The monthly exchange rate series for Cameroon and Malaysia, expressed in domestic currency per USD, are illustrated in Figure 2. Co-movements were observed for both series. Asia faced a financial crisis between September 1998 and July 2006. During this time period, Malaysia adopted a fixed exchange rate regime. The exchange rate was fixed at 3.8 MYR/USD. The French Devaluation of the African Currency (January 1994) explains the sharp increase in the Cameroon exchange rates in 1994 (Figure 2).

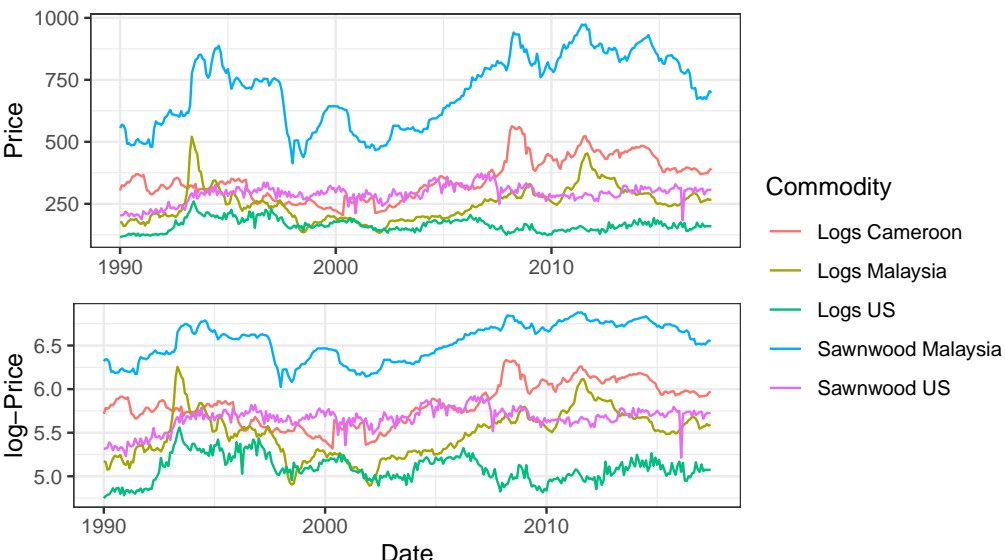

**Figure 1.** Price Series for Forest Products in Levels and in the Natural Logarithm in Cameroon, Malaysia and the US. The data covers a set of monthly average prices ranging from January 1990 to November 2019. The prices are expressed in US dollars per cubic meter. Foreign and domestic prices were collected from the United Nations Conference on Trade and Development (UNCTAD), IndexMundi and the World Bank.

We start the analysis by estimating the ERPT relationship via Equation (3). The results from a simple OLS estimation are displayed in Table 1. According to the results, the four log-prices are strongly and significantly dependent on the prices of the corresponding substitute commodities and exchange rates, which confirms the validity of ERPT in the analyzed markets. The only exception is the prices of Logs Malaysia/Logs USA, for which the exchange rate is found to be insignificant. The insignificance of the exchange rate may be due to non-linearities in the price relationship as we suggested. This result is in line with the previous works of Goodwin et al. [16]; Caselly and Roitman [14]; Wiseman, Luckstead and Durand–Morat [17]; Tao, Diao and Cheng [19]; Anderl and Caporale [11]; and Alola, Usman and Alola [13] that support the usage of non-linear models in investigating ERPT effects.

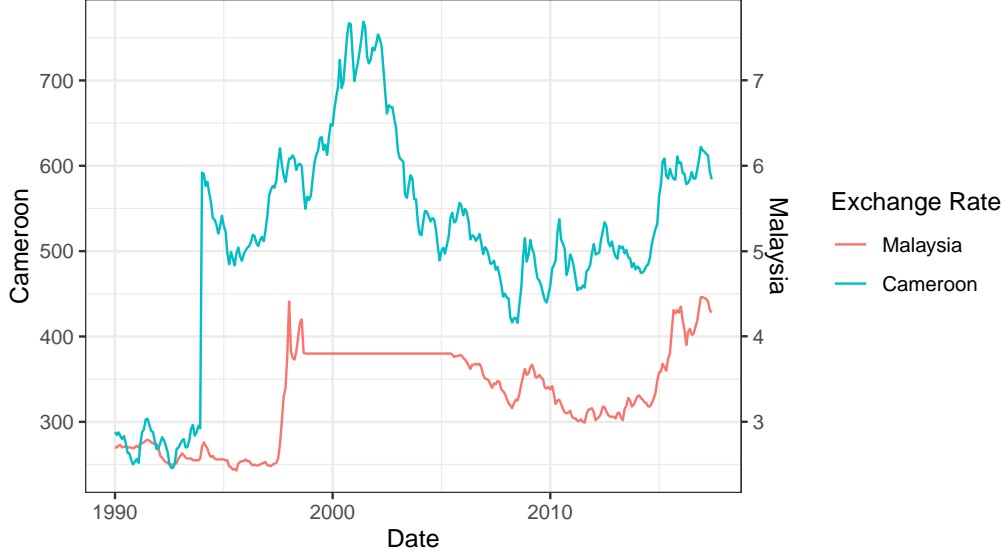

**Figure 2.** Nominal Exchange Rate, USD per Malaysian ringgit and USD per Central African CFA franc of Cameroon. The data covers a set of monthly average exchange rates ranging from January 1990 to November 2019. Exchange rate data were collected from the International Financial Statistics (IFS) curated by the International Monetary Fund (IMF).

**Table 1.** Exchange rate pass-through coefficient estimates.

|  | Logs Cameroon | Logs Cameroon | Logs Malaysia | Sawnwood Malaysia |
|---|---|---|---|---|
| **Prices** | | | | |
| Logs Malaysia | 0.145 *** | | | |
|  | (0.011) | | | |
| Logs USA | | 0.294 *** | 1.367 *** | |
|  | | (0.025) | (0.057) | |
| Sawnwood USA | | | | 0.424 *** |
|  | | | | (0.029) |
| **Exchange Rates** | | | | |
| Cameroon | 1.772 *** | 1.684 *** | | |
|  | (0.012) | (0.022) | | |
| Malaysia | | | −0.167 | 4.026 *** |
|  | | | (0.255) | (0.149) |
| $R^2$ | 0.999 | 0.999 | 0.972 | 0.991 |
| Adj. $R^2$ | 0.999 | 0.999 | 0.972 | 0.991 |
| Num. obs. | 330 | 330 | 330 | 330 |

The values in parentheses are the standard errors of the estimated coefficients. All data are in logarithms. The specification corresponds to Equation (3). *** $p < 0.01$.

The next step involves employing the most commonly used non-linearity tests: Teräsvirta [59], White [60], Keenan [61], McLeod–Li [62], Tsay [63] and the LR-Threshold [64] for each of the price ratios. The results provide evidence supporting non-linear processes (Table 2). This suggests that the ERPT effects may be regime-dependent. The stationarity of the price pairs were assessed using the Dickey–Fuller and Phillips–Perron tests (Table 3). Stationarity tests confirm that all price ratios are non-stationary with an order of integration 1 (i.e., $I(1)$). Hence, we estimate all models using the first-difference of the log–price ratios.

**Table 2.** Non-linearity tests for price ratios in first difference.

|  | Teräsvirta | White | Keenan | McLeod-Li | Tsay | LR-Threshold |
|---|---|---|---|---|---|---|
| Δ(Logs Cameroon/Logs Malaysia): | 1.559(2) | 0.335(2) | 2.364(2) |  | 1.447(2) | 744.280 |
|  | (0.459) | (0.846) | (0.125) | (0.044) | (0.229) | (0.000) |
| Δ(Logs Cameroon/Logs US): | 0.893(2) | 0.002(2) | 0.698(1) |  | 0.796(1) | −317.340 |
|  | (0.64) | (0.999) | (0.404) | (0.092) | (0.373) | (0.000) |
| Δ(Logs Malaysia/Logs US): | 1.904(2) | 0.233(2) | 0.001(8) |  | 1.039(8) | $2.28 \cdot 10^5$ |
|  | (0.386) | (0.89) | (0.977) | (0.001) | (0.414) | (0.000) |
| Δ(Sawnwood Malaysia/Sawndwood US): | 31.147(2) | 21.587(2) | 23.990(3) |  | 4.666(3) | −37.202 |
|  | (0.000) | (0.000) | (0.000) | (0.000) | (0.000) | (0.000) |

The prices are expressed in logarithms. Δ corresponds to the first difference operator. Teräsvirta [59] and White [60] test the null hypothesis that the linearity is in the mean. Keenan's one–degree [61], Tsay's [63] and likelihood ratio test [64] for threshold non-linearity, where the null hypothesis is that the time series follows the form of a two-regime SETAR process. McLeod–Li [62] test's null hypothesis is that the time series follows a form of the ARIMA process (maximum $p$–value reported.) The likelihood ratio test for the threshold non-linearity alternative hypothesis is that the time series follows a form Threshold Autoregressive Regression. The numbers in parentheses next to the statistics are the degrees of freedom or the test order.

**Table 3.** Unit root test results for forest products markets pass-through data.

|  | Level | | | | First Difference | | | |
|---|---|---|---|---|---|---|---|---|
|  | ADF Test | | PP Test | | ADF Test | | PP Test | |
| log–Ratio | Statistic | $p$-Value | Statistic | $p$-Value | Statistic | $p$-Value | Statistic | $p$-Value |
| Logs Cameroon/Logs Malaysia | −4.268 | <0.01 | −25.534 | 0.021 | −7.719 | <0.01 | −254.105 | <0.01 |
| Logs Cameroon/Logs US | −2.834 | 0.225 | −13.462 | 0.356 | −7.469 | <0.01 | −419.921 | <0.01 |
| Logs Malaysia/Logs US | −2.336 | 0.435 | −15.531 | 0.240 | −8.603 | <0.01 | −423.173 | <0.01 |
| Sawnwood Malaysia/Sand wood US | −1.835 | 0.646 | −19.754 | 0.074 | −8.718 | <0.01 | −414.075 | <0.01 |

The ADF Test column reports the Heteroskedasticity Robust Augmented Dickey–Fuller test statistics and the corresponding $p$-values. The PP Test column shows the Phillips–Perron unit root test statistics and $p$-values.

Given that the variables are non-stationary, Equation (3) should now be viewed as a cointegrating regression. This regression would reflect the long run relationships between the price and exchange rate variables [16]. Following Engle and Granger [65], we test possible cointegration relationships between the commodity prices and the exchange rates. The results are shown in Table 4. We find evidence of cointegration for two out of four commodity prices and exchange rates at the 1% significance level. This information is important for justifying the use of non-linear SETAR models to estimate the ERPT effects. These findings are similar to those of other authors: Goodwin et al. [49], Cheikh [32], Cheikh and Younes [28], and Anderl and Caporale [11].

**Table 4.** Engler–Granger cointegration test for forest products markets pass-through data.

| Price (Domestic/Foreign) | EG Statistic | $p$-Value |
|---|---|---|
| Logs Cameroon/Logs Malaysia | −3.652 | 0.029 |
| Logs Cameroon/Logs US | −1.885 | >0.100 |
| Logs Malaysia/Logs US | −3.399 | 0.047 |
| Sawnwood Malaysia/Sawnwood US | −1.802 | >0.100 |

The Engle–Granger alternative hypothesis test illustrates that the series are cointegrated. The Column EG statistic illustrates the unit root test of the residuals of an Engle–Granger cointegration regression of the domestic price on the foreign price and exchange rate. The corresponding $p$-values are interpolated from [66], Table 4.2, p.103. The lag length was selected as $k = \text{trunc}(n − 1)^{1/3} = 6$.

We use an agnostic approach to select the number of autoregressive lags in each regime. For each of the four log–price ratios, we estimate $2^3 = 8$ specifications of autoregressive processes up to the second lag in each regime. The three possible specifications for each

regime include the following: AR(0), $\phi_0^{R_k}$; AR(1), $\phi_0^{R_k} + \phi_1^{R_k} y_{t-1}$; and AR(2), $\phi_0^{R_k} + \phi_1^{R_k} y_{t-1} + \phi_2^{R_k} y_{t-2}$, where $k = 1$ and $k = 2$ represent regimes 1 and 2, respectively.

We then select the "best" model by considering both the number of statistically significant parameters and the Akaike [67], Bayesian [68], Shibatta [69] and Hannan–Quinn [70] information criteria (the full set of estimated models is available on request from the authors). All models are estimated by a maximum likelihood using the BFGS algorithm. This approach illustrates an improvement in fit, when compared with the typical one-lag approach. More specifically, out of four price pairs, only one has one lag in both regimes (Table 5, column (1)). We also find that the transition variable $\gamma$ is significant in most estimations, with only one exception. The threshold parameter $c$ is significant in all but one specification. These results support the necessity of modeling ERPT using flexible specifications that allow up to two lags, as well as the validity of the two-regime processes.

In our analysis, each model is denoted as a two-regime SETAR($p1, p2$), where $p_1$ and $p_2$ are the number of lags in the first and second autoregressive regimes, respectively, which can be 0 (constant only), 1 or 2. For example, by examining the third estimation (Table 5), the selected empirical estimation is a two-regime SETAR$(1, 2)$, with an AR(1) process in the first regime and an AR(2) process in the second regime:

$$y_t = (\phi_0^{R_1} + \phi_1^{R_1} y_{t-1}) \, F(y_{t-1}; \gamma, \alpha, c) + (\phi_0^{R_2} + \phi_1^{R_2} y_{t-1} + \phi_2^{R_2} y_{t-2}) \, (1 - F(y_{t-1}; \gamma, \alpha, c)) + \varepsilon_t \tag{7}$$

The persistence of the deviations from the LOP are characterized by non-linear adjustments. The adjustments are contingent upon the exchange rate passing the threshold value $c$. The threshold value may represent the transaction costs between regions in the forestry industry. The existence of the threshold makes the two-regime SETAR applicable to our case.

In this investigation, reasonable threshold values $c$ are estimated. This result concurs with the theoretical arguments in the international trade literature including Güney [45], Goodwin and Piggott [71], Bhat, Nain and Bhat [27] and Anderl and Caporale [11]. We observe that threshold values change across countries. We also observe heterogeneity in transaction costs for the same commodities (using USD as our reference currency). In part, this is due to the country-specific factors [72] such as movement restrictions, infrastructural barriers, differences in storage capacity and access to credit. For some commodities, different countries have higher thresholds than others.

Lastly, to assess whether or not shocks given to the price ratios have any important effects, the generalized impulse response functions are estimated. This is conducted by employing the methodology in Koop et al. [73] via bootstrapping with 500 replications. The generalized impulse response functions are provided in Figure 3. We add a positive and a negative impulse of one standard deviation of the prediction error. The results reveal that all impulses fade out, indicating non-permanent shocks with varying dynamics in terms of amplitude, oscillation and convergence time. Next, we proceed with a detailed analysis of each price pair followed by an overall conclusion regarding the markets and policy implications.

**Table 5.** Selected SETAR [1] models for forest products price ratios.

| Parameter | Logs Cameroon Logs Malaysia | Logs Cameroon Logs US | Logs Malaysia Logs US | Sawnwood Malaysia Sawnwood USA |
|---|---|---|---|---|
| | **(1)** | **(2)** | **(3)** | **(4)** |
| $\phi_0^{R_1}$ | 0.185 *** | 0.322 | 0.342 *** | −0.509 *** |
| | (0.062) | (1.232) | (0.059) | (0.088) |
| $\phi_1^{R_1}$ | −0.642 *** | 1.998 | −1.999 *** | . |
| | (0.237) | (5.367) | (0.171) | |
| $\phi_2^{R_1}$ | . | 22.130 ** | . | . |
| | | (9.592) | | |
| $\phi_0^{R_2}$ | −0.000 | 0.000 | 0.001 | 0.024 |
| | (0.003) | (0.004) | (0.004) | (0.069) |
| $\phi_1^{R_2}$ | 0.295 *** | −0.289 *** | −0.212 *** | −0.090 |
| | (0.076) | (0.062) | (0.059) | (0.359) |
| $\phi_2^{R_2}$ | . | −0.055 | 0.055 | . |
| | | (0.052) | (0.063) | |
| $\gamma$ | 980.666 *** | 61.754 *** | 75.261 *** | 10.282 |
| | (149.788) | (15.513) | (7.121) | (9.311) |
| $c$ | 0.139 *** | 0.348 *** | 0.333 *** | 0.321 *** |
| | (0.001) | (0.016) | (0.002) | (0.050) |
| $\alpha$ | 1.000 | 1.000 | 1.000 | 1.000 |
| $\sigma$ | 0.054 *** | 0.077 *** | 0.077 *** | 0.071 *** |
| | (0.004) | (0.004) | (0.004) | (0.006) |
| LogL | 492.273 | 374.745 | 377.554 | 405.540 |
| Akaike | −2.950 | −2.223 | −2.247 | −2.429 |
| Bayes | −2.869 | −2.120 | −2.154 | −2.360 |
| Shibatta | −2.951 | −2.225 | −2.248 | −2.429 |
| Hannan–Quinn | −2.918 | −2.182 | −2.210 | −2.401 |
| R-Squared | 0.091 | 0.117 | 0.081 | 0.221 |
| R-Squared (adj) | 0.072 | 0.092 | 0.058 | 0.207 |
| RSS | 0.966 | 1.974 | 1.941 | 1.637 |
| Skewness (res) | −0.043 | −0.055 | 0.067 | 1.446 |
| Ex.Kurtosis (res) | 5.116 | 1.293 | 0.944 | 8.806 |

[1] SETAR stands for Self-Exciting Threshold Autoregressive model. The estimated model is as follows: $y_t = (\phi_0^{R_1} + \phi_1^{R_1} y_{t-1} + \phi_2^{R_1} y_{t-2})\, F(y_{t-1}; \gamma, \alpha, c) + (\phi_0^{R_2} + \phi_1^{R_2} y_{t-1} + \phi_2^{R_2} y_{t-2})\,(1 - F(y_{t-1}; \gamma, \alpha, c)) + \varepsilon_t$, where the dependent variable $y_t$ is the logarithm of the price ratio at time $t$; $R_1$ and $R_2$ indicate the model parameters $\phi$ corresponding to the first and second regimes, respectively. $\varepsilon_t$ is a non-identical, independent random error, with mean zero and constant variance $\sigma^2$. $F(y_{t-1}; \gamma, \alpha, c)$ is the logistic transition function with the speed of the adjustment parameter $\gamma > 0$ and the threshold parameter $c$. For identification purposes, we set $\alpha = 1$. Robust standard errors are in parenthesis. A dot in the coefficients means that the corresponding autoregressive lag was not included in the specification. The information criteria are Akaike [67], Schwarz [68], Shibata [69] and Hannan and Quinn [70]. The values in parentheses are the standard errors of the estimated coefficients. *** $p < 0.01$; ** $p < 0.05$.

Table 5 illustrates that the estimated price ratios for logs in Cameroon conform to different variants of the SETAR setup: a SETAR(1, 1) for the ratio with Malaysia and a SETAR(2, 2) for the ratio with USA. Most parameters are statistically significant. Furthermore, the second-order autoregressive parameter is present and significant in the selected models, which proves the necessity of including the second-order lags.

The impulse response functions in Figure 3 show that a positive shock introduced into the system will initially lead to a decrease in the price ratios. The price ratios tend to converge to the equilibrium level at about 3.5 to 5 periods in an oscillating fashion between positive and negative values for the US and in a smoothly decreasing fashion for Malaysia. A similar path is observed for the negative shocks in terms of the time required for the price pairs to tend to converge to an equilibrium point.

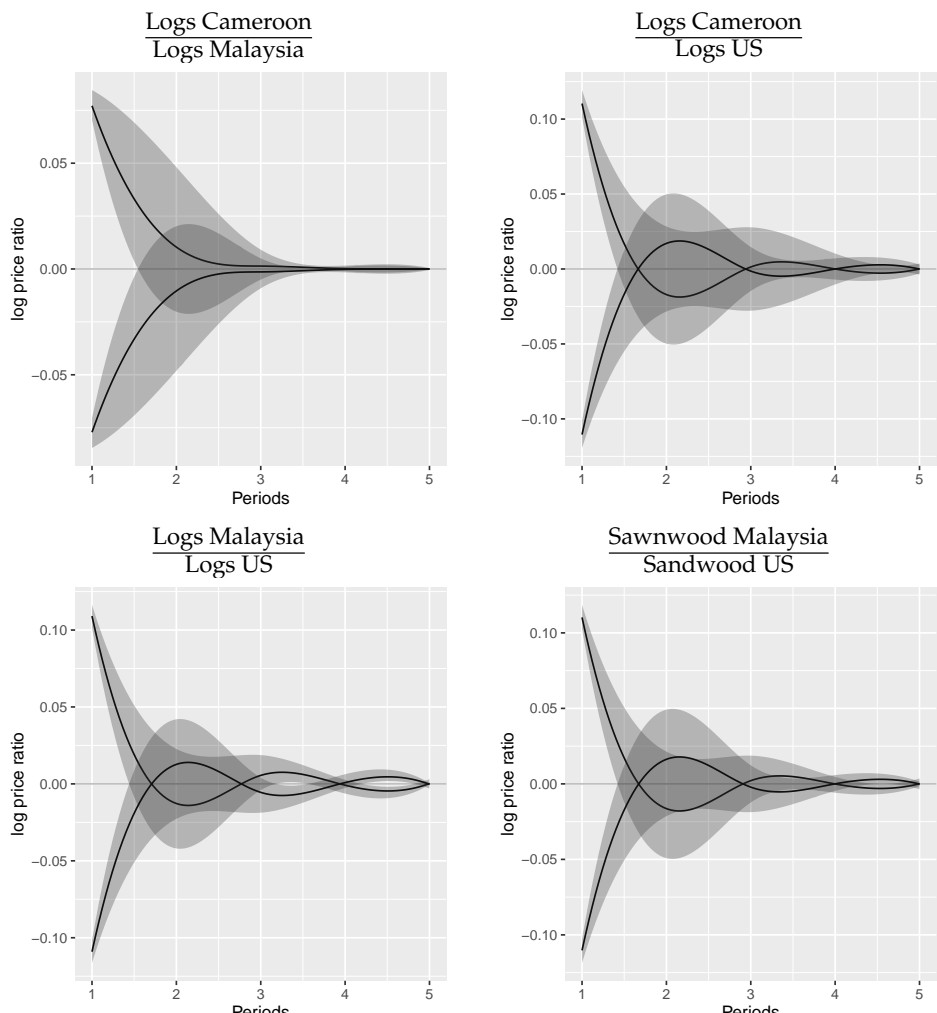

**Figure 3.** Impulse Response Functions for a One Standard Deviation Shock to the Monthly Forest Products Price Ratios. Each graph illustrates the 95% confidence interval of an impulse of $\pm 1$ standard deviation of the estimation error, based on 500 bootstrap repetitions.

The smoothing parameters suggest that the switching between regimes occurs at different speeds and in a smooth fashion. The value for the adjustment coefficient $\gamma$ between Cameroon and Malaysia is 980.666, whereas the $\gamma$ coefficient between Cameroon and the USA is 61.754. This shows that Cameroon has the highest variation in the switching speed between the two regimes. It should be noted that $\gamma$ is significant in both ratios.

The threshold parameters $c$ is 0.139 for the ratio Logs Cameroon/Logs Malaysia and 0.348 for Logs Cameroon/Logs USA. For each threshold parameter, we identify the first regime ($R_1$) as the one in which the percentage change of the price ratio is above $c$. In such a case, arbitrage is profitable and the process is mean reverting. On the other hand, the $R_2$ regime occurs when the deviations from the LOP are smaller than $c$. In the latter case, opportunities for profitable arbitrage are not observed. In addition, the parity condition will not be satisfied, as the exchange rate does not tend to converge to an equilibrium.

Generally speaking, based on the impulse response graphics in the Cameroon forest product markets, we observe important arbitrage opportunities for the US and Malaysia. This fact is supported by significant $c$ coefficients. The adjustment speeds of the prices differ across the regions, as evidenced by varying magnitudes of $\gamma$ coefficients. This could be due to the factors specific to the region that require further investigation.

The estimated model for the sawnwood market considers the Malaysia/USA ratio. The most suitable specification is a SETAR$(0, 2)$, with only a constant at the first regime being significant. The second-order autoregressive parameters are not useful in capturing the ERPT effects. For positive and negative shocks introduced into the system, the impulse response functions illustrate that the price ratios return to the equilibrium levels at about four to six periods in an oscillating way between the positive and negative values. However, since the smoothing parameter is not significant, we cannot make a conclusion regarding gains from arbitrage in these markets.

In summary, the significance of the estimated models and the impulse response graphics illustrate that Cameroon and the US can benefit from arbitrage with Malaysia in the logs markets. However, we do not observe any arbitrage opportunities between Malaysia and USA in the sawnwood market.

## 4. Discussion

In keeping with Ghalanos [74], this paper examines a regime-dependent ERPT for forest product prices (i.e., sawnwood, logs), by incorporating two-regime SETAR models in the US, Cameroon (West Africa) and Malaysia (Southeast Asia). Considerable volatility in the price ratios suggests the potential for significant market interactions between the countries. Our results illustrate evidence for the convenience of the two-regime SETAR models to analyze ERPT in a non-linear fashion for the forest product markets. All the estimated price ratios show a significant threshold parameter $c$. This illustrates strong evidence for the necessity of incorporating non-linear regime-dependent models in estimating ERPT.

The results exhibit that the forest markets are characterized by non-linearity and structural changes. Theoretical price parity relationships are supported by the empirical application. We find feasible estimates for $c$ that may be regarded as transaction costs. This corresponds with the previous trade literature. Countries exhibit different threshold values. Overall, our analysis suggests differences in the potential arbitrage opportunities between countries.

The threshold values also provide information about the trade situations in each country. For instance, in Cameroon's logs market, if the increase in the lagged percentage change in log prices is higher than 35 percent in absolute value, the country will have potential gains from arbitrage with the US. Malaysia has this potential when the lagged percentage change in prices is higher than 33 percent. The results do not exhibit any evidence of arbitrage opportunities between Malaysia and USA in the sawnwood market.

Finally, the impulse response analysis for each price pair supports the changing behavior of the price ratios in different regimes. This serves as another justification for employing models that consider structural changes in a non-linear fashion to investigate the deviations from the parity conditions. We find that the adjustment processes after shocks are similar, ranging from 3.5 to 6 months. However, the amplitude of the impact differs among markets.

The results suggest the suitability of integrating the second-order lag into the SETAR framework. These results can be viewed as an improvement upon the previous literature using smooth transition autoregressive models to test the ERPT effects. The opportunities for profitable arbitrage and the different speeds of market adjustments in the forestry markets should be taken into consideration by the policymakers and traders in their decision-making processes.

## 5. Conclusions

This study focused on determining the responsiveness of international forest product prices to exchange rate variations and testing whether or not the differences in profitable arbitrage opportunities exist between the selected countries. The evidence obtained from non-linearity tests and the significance of the threshold parameters reveal that non-linearity and structural changes are important features to consider while investigating the validity of ERPT effects in the forest markets in Cameroon, Malaysia and the US. The responsiveness of

international forest product prices to variations in exchange rates differs between countries. We also observe differences in profitable arbitrage opportunities.

Our findings concur with the results in Uusivuori and Buongiorno [75] and in Goodwin et al. [16]. Our primary contribution is to improve upon traditional SETAR models by considering the use of autoregressive first-order dynamics in the state equations via including one lag of the dependent variable as the threshold variable. This results in better forecasts, since integrating more lags helps to capture the cumulative effects of the price dynamics. Our approach is more flexible compared with integrating two regimes with one lag in each regime. We incorporate up to two lags as well as different numbers of lags in each regime.

*Limitations and Future Direction*

This paper contributes to the time series literature by focusing on ERPT in the forestry industry at the commodity level. However, this article also has limitations. One limitation is that a minimal amount of information was provided by the lagged values of the dependent variables. Hence, we expect more efficient forecasts to come from the use of a set of independent explanatory variables. This paper also only considers one switching variable in the transition function. Future research may benefit from using a linear combination of independent variables in the regime equations and integrating multiple switching variables in the state dynamics. Furthermore, adding more regimes and integrating more lags to capture the cumulative effects may improve the accuracy of the estimates. Moreover, it might be useful to integrate macroeconomic measures such as the GDP, employment and net trade situation of the countries to investigate the ERPT effects. Additionally, future research may include explicitly investigating iceberg transport costs to elaborate the effects of distances between regions.

**Author Contributions:** Conceptualization, S.G. and B.G.; methodology, S.G. and B.G.; software, S.G. and A.R.; validation, S.G. and A.R.; formal analysis, S.G. and A.R.; writing—original draft preparation, S.G., A.R. and B.G.; visualization, S.G. and A.R. All authors have read and agreed to the published version of the manuscript.

**Funding:** This research received no external funding.

**Institutional Review Board Statement:** Not applicable.

**Informed Consent Statement:** Not applicable.

**Data Availability Statement:** Exchange rates source: IMF, Price data source: World Bank, Index Mundi and UNCTAD.

**Conflicts of Interest:** The authors declare no conflict of interest.

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
