# Peer review of "An Analysis of the Pass-Through of Exchange Rates in Forest Product Markets"

_agriculture, doi:10.3390/agriculture13030515_

Round 1

Reviewer 1 Report

P. 2. Line 40: This link seems outdated. Maybe this: https://apps.fas.usda.gov/gats/default.aspx

P. 8. Line 170: Have you considered the “Iceberg Transport Cost” (Samuelson, 1954) problem or is it not a problem in this case since you refer to prices? Also, the weight of wood can change while transporting across long distances and the accounting year can be revised. Is this raw data or has it been previously adjusted?

P. 10 Line 213: Can you explain the insignificance here?

P. 10 Line 226-229: Sawnwood is not significant. Could this mean that the model is not suitable for this product category?

P. 11 Line 235-244: In my opinion, it would be more comprehensible for the reader to state which variables are insignificant, instead of having to look it up in the table (also in other parts of the text). However, maybe this will improve, when the table is closer to the text.

P. 16 Line 365: Consider my comment about line 170, if relevant.

Figure 1: What prices are these? From my understanding of how you explained ERPT, it should be import prices. Consider adding this, if not obvious. When you refer to domestic, do you mean US and foreign outside US?

In general:

The manuscript is well written and structured.

Do you plan to include your calculations in the final publication? This would be very helpful for reproducible results and further understanding.

Author Response

We highly appreciate your insightful and helpful comments on our manuscript.

Please find the detailed responses to your reviews and recommendations below:

Some of your suggestions involved considering the “Iceberg Transport Cost” in our analysis or including it for future work. Iceberg transport costs are implicitly included as a part of the transaction’s costs. Investigating the iceberg costs explicitly is out of the scope of this paper, so we have added the concept to be investigated for future work. We agree that this is a very interesting avenue for future inquiry.

You asked for more clarification on our data, asking if this is raw data or if it has been previously adjusted.  We have included the measurement unit of the prices (dollar per cubic meter) and indicated that this is raw data. Also, we were asked to be more specific on what type of prices were used in the analysis, import vs. export prices. We indicated that import prices were used and added this information as a footnote in Figure 1.

We were asked to clarify what we refer to as domestic prices, if we mean US and foreign outside US. We agree with your review again, and we fixed the footnote of Figure 1. Based on your comment, we replaced the original wordings by “prices”.

You also requested us to explain the insignificance of the exchange rate between US and Malaysia. We have added the following information on why this might be the case: “LR–Threshold value from Table 2 reveals significant nonlinear relationship between US and Malaysia for logs, for that reason the ERPT effect could not be captured by a linear model which might the be reason for this insignificance.

Also, you asked to clarify if the model is not suitable for sawnwood product category as it had an insignificant coefficient in the model. We agree with you, and we have mentioned that the results do not exhibit any evidence of arbitrage opportunities between Malaysia and the US in the sawnwood market, which implies this model may not be the best model for this product category.

Additionally, you provided recommendations on improving the exposition of our paper. It was stated that it would be more comprehensible for the reader to state which variables are insignificant, instead of having to look it up in the table (also in other parts of the text) and stated that when the table is closer to the text the problem might be solved. We have put the tables closer to the text to make it more convenient for the readers.

You warned us about a link that seems outdated. We updated the link to https://apps.fas.usda.gov/ gats/default.aspx

Lastly, you asked if we plan to include our calculations in the final publication. The calculations will be available based on request.

We are very grateful for your willingness to consider a revision. 

We have attempted to address each of your comment and suggestion with edits to the manuscript and revisions to the article. While we hope that we have sufficiently addressed all your concerns, we stand ready to make additional edits in accordance with your suggestions or to respond to any remaining concerns.

Best Regards

Selin Guney

Reviewer 2 Report

When the article is examined there are some issues that need to be corrected.  The following details my specific comments on the paper.

1.     It is seen that the authors do not comply with the instructions for the authors. Main text, tables, figures and references do not comply with journal editorial rules. Authors are advised to follow the instructions of journal.

2.     The discussion section of the article should be expanded a little more. In the meantime, the relevant literature should be used.

3.     Line 121: This investigation study also contributes to………

Author Response

We highly appreciate your insightful and helpful comments on our manuscript.
Please find the detailed responses to your reviews and recommendations below:

You indicated that we do not comply with the instructions for the authors. We completely agree with this and we have fixed the main text, tables, figures and references to comply with journal editorial rules.

We were recommended to expand the discussion section of the article a little bit more with the relevant literature being used. The relevant latest literature was integrated into the paper. Twelve recent papers that considered ERPT were added to the paper and our results were discussed and compared with the findings of the aforementioned articles where applicable in the results section. We have added the discussion to the results section where we thought would be more appropriate for the flow of the paper. That said, we would be happy to pursue other alternatives at your recommendation. Also, we have updated the information in the introduction part using the most updated reports and data webpages.

You referred to a particular line (Line 121) for the grammar of the sentence to be fixed. We have fixed the sentence according to your recommendation.

We are very grateful for your willingness to consider a revision. 
We have attempted to address each of your comment and suggestion with edits to the manuscript and revisions to the article. While we hope that we have sufficiently addressed all your concerns, we stand ready to make additional edits in accordance with your suggestions or to respond to any remaining concerns.
Best Regards
Selin Guney

Reviewer 3 Report

The introduction part must b improved with the latest papers. 

Methods need to improve, and robustness is missing. 

The results section needs to be improved and discuss in detail with the latest papers. 

The contribution of the study is too short.

Limitations and future direction must be separate. 

Author Response

We highly appreciate your insightful and helpful comments on our manuscript.
Please find the detailed responses to your reviews and recommendations below:

Based on your recommendation, the introduction part has been updated with the latest available data and papers (where applicable).

You indicated that methods need to improve and mentioned that robustness checks are missing.

The paper employs one of the latest nonlinear modelling technique, which has been successfully used to investigate the ERPT effects in many sectors before. This successful methodology has been further extended in our paper the forestry sector, which is one of the main contributions of this paper. Our results include the estimation of 8 different specifications for the 4 selected price ratios, which confirms robustness. For the space saving purposes we presented the best fit models in the paper. The remaining estimations can be provided upon request.

You recommended that the results section needs to be improved and discussed in detail with the latest papers. In an aim to do so, we have added the latest literature to our paper. 12 recent papers that considered ERPT were added to the paper and our results were compared with the findings of the new articles where applicable. For the methodology references, we kept the original papers where the models were first introduced, not the papers that used these methodologies in the latest literature.

You suggested to include the limitations and future direction of the study in a separate section. We have formed a separate section to discuss the limitations of the study and recommendations for future research complying with the journal’s guidelines. 

You indicated that the contribution of the study is too short. This paper has two main contributions to the literature. Our primary contribution is to improve upon traditional SETAR models by considering the use of autoregressive first order dynamics in the state equations via including one lag of the dependent variable as the threshold variable. This results in better forecasts since integrating more lags helps to capture the cumulative effects of the price dynamics. Our approach is more flexible compared to integrating two regimes with one lag in each regime. We incorporate up to two lags as well as different numbers of lags in each regime. To our knowledge this is the first study that extends upon traditional SETAR models to include different numbers of lags in each regime.

This paper also contributes to the time series literature by focusing on ERPT in the forestry industry at the commodity level. There are only few articles published in the forestry sector considering ERPT effects on the commodity level.

We are very grateful for your willingness to consider a revision. 
We have attempted to address each of your comment and suggestion with edits to the manuscript and revisions to the article. While we hope that we have sufficiently addressed all your concerns, we stand ready to make additional edits in accordance with your suggestions or to respond to any remaining concerns.
Best Regards
Selin Guney